# The Prevalence and Determinants of Vitamin D Status in Community-Dwelling Older Adults: Results from the English Longitudinal Study of Ageing (ELSA)

**DOI:** 10.3390/nu11061253

**Published:** 2019-06-01

**Authors:** Niamh Aspell, Eamon Laird, Martin Healy, Tom Shannon, Brian Lawlor, Maria O’Sullivan

**Affiliations:** 1School of Medicine, Trinity College, Dublin 2, Ireland; niamha@tcd.ie (N.A.); lairdea@tcd.ie (E.L.); 2Department of Biochemistry, St James’s Hospital, Dublin 8, Ireland; healymartinj@gmail.com; 3MedlLab Pathology Dublin 8, Ireland; thomasshannon2@gmail.com; 4School of Medical Gerontology and Institute of Neuroscience, Trinity College, Dublin 2, Ireland; lawlorba@tcd.ie

**Keywords:** vitamin D, 25-hydroxyvitamin D, latitude, deficiency, community-dwelling

## Abstract

Vitamin D deficiency is often associated with adverse health outcomes in older adults. The circulating 25-hydroxyvitamin D (25(OH)D) status predominately relies on UV exposure. However, the extent of which northerly latitude exasperates deficiency is less explored in ageing. We aimed to investigate vitamin D deficiency in community-dwelling, older adults, residing at latitudes 50–55° north. This study was comprised of 6004 adults, aged >50 years from wave 6 (2012–2013) of the English Longitudinal Study of Ageing (ELSA). Deficiency was categorised by two criteria: Institute of Medicine (IOM) (<30 nmol/L) and Endocrine Society (ES) (<50 nmol/L). The overall prevalence of Institute of Medicine (IOM) and Endocrine Society (ES) definitions of deficiency were 26.4% and 58.7%, respectively. Females (odds ratio (OR) 1.23; CI: 1.04–1.44), those aged 80+ (OR: 1.42; CI: 1.01–1.93), smoking (OR: 1.88; CI: 1.51–2.34); of non-white ethnicity (OR: 3.8; CI:2.39–6.05); being obese (OR: 1.32; CI:1.09–1.58), and of poor self-reported health (OR:1.99; CI:1.33, 2.96), were more likely to be vitamin D deficient (by IOM). Residents in the south of England had a reduced risk of deficiency (OR: 0.78; CI:0.64–0.95), even after adjustment for socioeconomic and traditional predictors (obesity, age, lifestyle, etc.) of vitamin D status. Other factors, such as being retired, having a normal BMI, engaging in regular vigorous physical activity, vitamin D supplement use, sun travel, and summer season were also significantly positive correlates of deficiency. Similar results were observed for the ES cut-off definition. Importantly, more than half of adults aged >50 years had 25(OH)D concentrations <50 nmol/L. These findings demonstrate that low vitamin D status is highly prevalent in older English adults and the crucial importance of public health strategies throughout midlife and older age to achieve optimal vitamin D status.

## 1. Introduction

By 2030, the number of adults aged 60 years and older is projected to increase by 56% and to double again by 2050 [1]. However, whilst we are achieving greater longevity and increasing good health into old age, this is not the case for all. In response to the age-related demographic changes, healthcare strategies that promote successful ageing through targeting modifiable lifestyle and dietary risk factors are of the utmost importance [2]. Moderate physical activity, a high-quality diet, and maintaining a normal body weight all contribute to successful ageing, with emerging evidence for dietary components, such as vitamin D [3,4,5,6]. 

Vitamin D is a steroid hormone derived primarily from sunlight exposure and to a lesser extent from diet [7]. Several known age-related adverse health outcomes, including bone loss, fracture risk, and falls, are associated [8,9,10] with suboptimal serum 25-hydroxyvitamin D (25(OH)D) status (the main circulating and diagnostic metabolite of vitamin D) with further evidence for non-skeletal roles in carcinogenesis, immune function, cardiovascular disease, dementia, and all-cause mortality [11,12]. Cross-sectional studies demonstrate that 25(OH)D concentrations tend to be higher from teenage life to about 65 years of age and then are lower again in older age [13], largely attributable to physiological, social, and lifestyle changes [14]. Despite increased emphasis and initiatives to optimise vitamin D status in older populations [15], the prevalence of deficiency continues to be of concern [16]. The reported prevalence of vitamin D deficiency in older populations across Europe and the UK vary considerably, with studies varying between 0–77% [17,18,19]. Vitamin D deficiency in older adults is multifactorial, influenced by season, supplement use, age-related behavioral changes including clothing and less time spent outdoors, health status, and age-related changes in vitamin D metabolism [7]. Evidence supports the effectiveness of vitamin D food fortification on winter serum 25(OH)D concentrations [20]. However, the uptake of both fortified foods and vitamin D containing supplements remains low in older adults in the UK. In Northern Europe, countries of latitudes greater than 40° N (including England at 50–55° N) have inadequate ultra-violet B light (UVB) exposure for vitamin D synthesis for 6 months of the year (October to March) [21]. However, little is known about the influence of regional geographic location and demographics on vitamin D status in older English adults.

In the present study, we investigated vitamin D status in a large population of English community-dwelling adults aged ≥50 years The specific aims of this study were; (i) to establish the prevalence of serum 25(OH)D deficiency in older adults, (ii) to determine the prevalence of deficiency by region of residence and, (iii) to identify the potential determinants of deficiency. 

## 2. Materials and Methods

### 2.1. Study Design and Population

The current study includes participants from the English Longitudinal Study of Ageing (ELSA), which is an ongoing nationally representative study of health. ELSA consists of men and women born on or after the 29 February 1952. The Health Survey of England recruited participants using multistage stratified probability sampling with postcode sectors selected initially and household addresses thereafter. The ELSA study sample was selected from the Health Survey for England (HSE) sample in 1998, 1999, and 2001, and a biennial follow-up is ongoing since. Further details of ELSA have been reported previously [22]. For the present analysis, cross-sectional data from wave 6 were used (2012–2013), as this was the first time 25(OH)D concentrations were collected for the new ELSA study sample. A total of 10,601 subjects were interviewed at wave 6. For the present analysis, participants were included if they were aged ≥50 years old, identified as a resident in England, and had a measured concentration of 25(OH)D; the latter required a blood sample to be collected during the nurse visit stage. After excluding those who did not complete the core and nurses’ data collection at wave 6, or had missing data for other key variables, the final analytical sample was comprised of 6004 subjects (Figure 1). Participants excluded due to age (<50 years (*n* = 65)) were typically Partners of the core members, and due to location (*n* = 21) were resident outside of England, typically Scotland and Wales. 

The ELSA study was conducted according to the guidelines laid down in the Declaration of Helsinki, and all procedures involving human subjects/patients were approved by the London Multicentre Research Ethics Committee (MREC 01/2/91). Written informed consent was obtained from all participants.

### 2.2. Study Measures

Serum 25(OH)D was assessed from a fasting blood sample collected during the nurses’ visit of wave 6 (all blood sampling occurred from January 2012 to July 2013). The laboratory analysis was undertaken at the Royal Victoria Infirmary (Newcastle upon Tyne, UK). Concentrations of 25(OH)D were analysed using Diasorin Liaison 25-hydroxyvitamin D immunoassay (which yields a lower detection limit of 3.0 nmol/L), and all assays were performed in duplicate (CV values ranged from 8.7% to 9.4%). The laboratory also participated in the Vitamin D External Quality Assessment Scheme (DEQAS). Vitamin D deficiency was defined according to two criteria, the Institute of Medicine's (IOM) guidelines for 25(OH)D concentrations (<30 nmol/L) [23], and the Endocrine Society (ES) cut-off (< 50nmol/L) [24], to allow for comparability with other reported prevalence data globally. Extreme outliers were defined as those with 25(OH)D >250 nmol/L, though none were identified in the population. Season was categorised using the extended vitamin D calendar for summer (April–Sept) and winter (Oct–March). Other health behaviours that could affect 25(OH)D status were available and included—sun holiday travel within the past 12 months (yes/no) and supplement use, defined as prescribed vitamin D and calcium required for treatment of osteoporosis (yes/no).

### 2.3. Classification by Region of Residence and Latitude

To protect the anonymity of the research participants, detailed postcode information was not available for this analysis. However, Government Offices for the Region (GOR) data were available for each participant, established by the national archives, which is regarded as the primary classification for the presentation of regional statistics. Each participant has been classified by the GOR for their location, to which we applied a central-most point of latitude, established by corresponding latitudes from weather stations by the Centre for Environmental Data Analysis (CEDA) and Google Earth. England ranges in latitude from the Lizard point, as the most Southerly point on the mainland (49°57′30″ N), to the Marshall Meadows Bay, as the most Northerly (55°48′18.3″). The potential maximal distance from the central point within each region was 1°. GOR divides England into nine regions. For the present study, regions were combined to compare between the North, the Midlands, and the South. Those residing in the North East, North West, and Yorkshire and The Hummer were defined as the North; those in East Midlands, West Midlands, and East England were defined as the Midlands; and those in the South East, South West, and London were defined as the South. Those identified as residing in Scotland, Wales, and Northern Ireland were not included in this analysis (*n* = 21).

### 2.4. Covariates

From the Wave 6 data, we identified important demographic, socioeconomic, lifestyle, and clinical contributors relevant to vitamin D status, with older adults and location of residence as covariates in the analyses. Age was recorded as a continuous number until 90 years (with ages above 90 collapsed to the value of 91 to ensure anonymity). Age category was also considered an important variable (age ≥60 years, retirement age (≥65 years)), sex, marital status (married, single never married, divorced/separated, or widowed), education level obtained (minimum o-level, i.e., Secondary school exam, or non-UK equivalent), including foreign equivalent), and employment status (employed, unemployed, retired) were considered important socio-demographic factors. Ethnicity was considered. However, 99% were white Caucasians. Health behaviours related to alcohol consumption (within the last 12 months (categorised to less than daily; daily, and times per week)); smoking status (current smoker, past smoker, never smoked); subjective physical activity levels (sedentary; low; moderate; and vigorous, with ≥1 time per week); and polypharmacy, defined as the use of ≥5 daily medications and self-reported health (poor, fair, good, very good, and excellent). Whether the participant had a long-standing illness, which was self-reported as limiting, was also considered. Anthropometric measures included BMI, calculated using the standard formula (kg/m²) from weight and height via nurse-obtained measures

### 2.5. Statistics

Descriptive statistics are presented as means (±SD), or proportions, for demographic and lifestyle characteristics. Overall prevalence figures are presented as crude and weighted results. The weighting strategy aimed to minimise any bias from a differential non-response in obtaining a blood sample. The core (analytical) sample for 25OHD is based on ELSA participants who took part in the nurse visit, and thus had a blood sample. Response to the blood sample was modelled on a range of household- and individual-level information collected from the ELSA wave 6 main interviews. Significant differences in participant characteristics were noted between those who engaged in the nurse visits and subsequently provided a blood sample, compared to those who did not respond to a nurse’s visit. To bring the characteristics more in line with what is known about the study population, available cross-sectional blood weights were applied to the multivariable logistic regression. The final blood sample weight was a product of this non-response weight and the wave 6 nurse weight. Detailed descriptions of the weighting principles applied by the ELSA Study can be found in the technical reports available at www.ifs.org.uk/elsa.

Differences between the groups were examined by independent Student t-tests, χ2, or ANOVA as appropriate. Weighted multiple logistic regression analysis was used to explore the determinants of 25OHD deficiency and presented as an odds ratios (OR) and 95% confidence intervals (CIs). We applied multivariate analysis of variance (MANOVA) to all explanatory variables to test the assumptions of the model, before including them in a stepwise regression. Only variables that reached statistical significance in the initial analysis (Appendix A and Table 3) were considered for the models, and, after testing for collinearity by scatter plots and Variance Inflation Factors Tests for linear relationships, those tests were performed by the inclusion of continuous predictor variables to the model and were reported as OR for a single unit change with 95% CI. In the adjusted models, covariates were selected based on previous research findings and following stepwise regression to test the contribution of each predictor (excluded *p* = 0.2500). The contribution of covariates to deficiency was examined by including each set of covariates as models in the following order: Model 1—vitamin D deficiency defined as <30 nmol/L. Variables included socio-demographics (age in years, sex, and region of residence), health-related behaviours (current smoker, vitamin D supplement use), physical factors (vigorous PA >1/week, BMI), and season (winter/summer blood sampling). In model 2, vitamin D deficiency was defined as <50 nmol/L and applied the variable selection of model 1. Corresponding unadjusted results are also reported. All analyses were performed in STATA 14.0 (StataCorp LP).

## 3. Results

A total of 6004 individuals aged 50 years or older were included in this study. Characteristics of the study population overall, stratified by vitamin D deficiency and region of residence, are detailed in Appendix A and Table 1. The mean sample age was 66.4 ± 8.8 years, with those aged 60–69 representing the largest age category (41.0%). Overall, 55% were female and most (73%) had achieved a minimum O-level education or equivalent. The majority (76.4%) rated their health as good or better than good, a quarter (25.7%) reported the use of five or more medications daily, and 31% reported a longstanding limiting illness. Consumption of alcohol at least once per week was estimated at 63.9%, and current smoking status was at 12%. An overweight and obese BMI was prevalent at 42.3% and 29.7%, respectively. Few study participants (4.4%) reported taking a prescribed vitamin D supplement, as indicated for osteoporosis treatment. Use of non-prescribed vitamin D containing supplements or dietary vitamin D intake was not available in the dataset. Characteristics of the cohort by region were significantly different for both educational attainment and health indicators. There were no statistically significant regional differences for vitamin D factors, namely season of blood draw, sun holiday travel, or prescribed vitamin D supplement use.

Overall, the prevalence of deficiency (<30 nmol/L and <50 nmol/L) was 23.7% (weighted 26.4%) and 55.3% (weighted 57.3%), respectively (Table 2). For older age categories of ≥60 years and ≥65 years, vitamin D deficiency (<30 nmol/L) and deficiency (<50 nmol/L) were similar at 22.6% and 22.5% (weighted: 25.3% and 25.3%) and 54.2% and 53.9% (weighted: 56.1% and 55.9%), respectively. The highest prevalence of deficiency (by IOM) was noted in women aged 80 years and older (36.8%), with the lowest in men aged between 70–79 years (17.3%). The overall mean 25(OH)D was 48.7 ± 23.5 (range; 9–239 nmol/L). Further demographic details of demographics by vitamin D status is presented in Appendix A. 

The prevalence of deficiency by geographic region and season are presented in Appendix A and Figure 2 and Figure 3. A difference in 25(OH)D concentrations and for the prevalence of deficiency by the north–south gradient was observed. The frequency of deficiency was significantly higher in residents of the North (*p* < 0.004) and the Midlands (*p* < 0.001) compared to those from the South of England. In line with this, mean serum 25(OH)D was also significantly lower in the North (*p* < 0.001), and the Midlands (*p* < 0.002) compared to the south. Similarly, when examined by latitude, mean 25(OH)D increased with decreasing latitude. Data from London, however, did not follow this pattern, showing higher deficiency and, correspondingly, lower serum 25(OH)D than comparable areas in the South. The highest prevalence of vitamin D deficiency was identified in the North East region during winter (36.4%), and the lowest in the South West was in summer (10.3%). Similarly, vitamin D deficiency (ES <50 nmol/L) was highest in the North East in winter (70.3%), though it remained high at 53.2% during summer. The lowest prevalence (34.9%) of deficiency (ES) was noted in the South West during summer. The prevalence increased from the most Southerly region (50.4° N) to the most Northerly (54.9° N) by a similar magnitude of ~10%, irrespective of the season.

After adjusting for pre-defined covariates. the multivariable logistic regression analysis revealed that being aged 80+, female, of non-white ethnicity, widowed, obese, a current smoker, and self-reporting fair or poor health were statistically significant negative correlates for vitamin D deficiency (<30 nmol/L), as detailed in Appendix A and Table 3. Factors such as being retired, having a normal BMI, engaging in regular vigorous physical activity, vitamin D supplement use, sun travel, and summer season were significant positive correlates of vitamin D deficiency. Similar variables contributed to the model for vitamin D deficiency (ES < 50 nmol/L), as shown in Appendix A, apart from female sex, which was no longer a statistically significant determinant. Among the independent correlates, non-white ethnicity was the strongest indicator of vitamin D deficiency (ES < 50 nmol/L), compared with Caucasians (OR 4.67; CI 2.57–8.51). Residing in the south of England was associated with a 22.8% lower risk of deficiency (IOM, <30 nmol/L) (OR 0.78; CI 0.64–0.95) (*p* = 0.003)). For each 1 degree increase in latitude, individuals were 11% more likely to be vitamin D deficient (IOM < 30 nmol/L) (*p* < 0.001) and 9% were more likely to be vitamin D deficient (ES < 50 nmol/L) (*p* < 0.001). 

## 4. Discussion

In this, the largest nationally representative study of vitamin D status yet conducted in community-dwelling older adults in England and Europe, we found that year-round vitamin D deficiency, irrespective of definition, was common. The prevalence of vitamin D deficiency was 26.4% (1 in 4) by IOM criteria, rising to 57.3% by ES criteria. The highest prevalence of deficiency was in females aged 80 years or over (36.8%). Important determinants of vitamin D deficiency (< 30 nmol/L) were being female, older age, widowhood, poorer self-rated health, and non-Caucasian ethnicity. Established risk factors for deficiency, such as obesity, smoking, and wintertime were also replicated. Prescribed vitamin D supplement use for bone health, though low, had protective effects, as did sun travel/holiday. The study also identified a north-south gradient, and incremental changes in latitude of residence influenced serum 25OHD concentrations, even within a country with a short-ranging latitude (50.4–54.9° N).

The prevalence of vitamin D deficiency observed in ELSA shows a similar deficiency pattern as the Longitudinal Ageing Study Amsterdam 17.5% (<30 nmol/L) and 48.4% (<50 nmol/L) [25]. Recent analysis of The Irish Longitudinal Study of Ageing (TILDA) reported the prevalence of vitamin D deficiency (<30 nmol/L) was 13.1% among Caucasian adults aged ≥50 yrs [17]. Previous figures from the UK National Diet and Nutrition Survey (NDNS) in 1995 reported vitamin D deficiency (using a lower cut-off of <25 nmol/L) was 19% for adults aged over 65 years [26]. The Health Survey England, also using <25 nmol/L deficiency criteria, reported a deficiency of 15% in 2000, increasing to 21% in 2005 [22]. For comparison, in the present study, when applying a lower cut-off of 25OHD <25 nmol/L, specifically to those aged 65 years and older only, 15% were deficient. Consistent with our data, these studies identified an increased risk of vitamin D deficiency in the oldest participants (≥80 years) Taken together, irrespective of criteria, the combined evidence highlights that a suboptimal 25(OH)D status is common in ageing populations. 

We have identified several important determinants of vitamin D deficiency in this older adult population. Consistent with the literature, smoking, obesity and lower physical activity were all negative determinants of 25(OH)D status [17,22,27,28,29]. Having a normal BMI was also associated with a 20% lower risk of vitamin D deficiency, irrespective of defining criteria, which has been reported previously [27,30]. The use of a prescribed vitamin D supplement for bone health was associated, with a reduced risk of vitamin D deficiency at 30 nmol/L. However, prescription vitamin D-containing supplement use was low at 4.4%, though results from Laird et al. [17] showed than even a relatively low usage of vitamin D supplements (8.5%, which included prescribed and non-prescribed supplements) protected against vitamin D deficiency. The uptake of vitamin D supplements, however, is generally low in community dwelling older populations [31]. This study also demonstrated a distinct north-south gradient in deficiency and in 25(OH)D concentrations, with those residing in the south of England identified as less likely to be vitamin D deficient irrespective of cut-offs applied. Latitude contributed significantly to 25(OH)D concentration, even within a country of such a short ranging latitude, with each 1° northerly increase associated with an 11% increased risk of vitamin D deficiency. Some of these observations could be explained by the reduced UVB exposure in the most Northern and Western parts of the Country [21], while lower socio-economic status (which may be more common in these areas) is a known risk factor for having a lower vitamin D level [32]. These regional differences have been observed in other countries, such as Ireland, which has a much smaller geographic area than England. In Ireland, stark regional differences were still observed [17]. Interestingly, even though London is located in the South of England, it experienced high levels of vitamin D deficiency. This finding may reflect urban lifestyles, the heterogeneous population (with extremes of affluence and poverty), or population aggregation, where more people with chronic conditions may aggregate in cities. The more densely packed housing and the ‘food-desert hypothesis’ (referring to deprived, populated urban areas where residents do not have access to an affordable and healthy diet, e.g., a lack of access to vitamin D rich foods) could account for this result, along with the effects of city living [32]. However, it must be noted that at the moment, the UK does not have a mandatory vitamin D food fortification policy, which could be contributing to the high levels of deficiency observed throughout the country. 

This study has a number of strengths and weaknesses. We were able to report the most up to date prevalence and determinants of vitamin D deficiency from a large, robust, nationally representative study of community-dwelling adults, with comprehensive health, demographic, and lifestyle variables. The ELSA study followed standardised protocols and complied with international quality control schemes to measure 25(OH)D concentration. However, the method used was the DiaSoirin Liaison immunoassay, which is subject to its own limitations, and we did not have access to the gold standard method of liquid chromatography mass spectrometry. Furthermore, details on sunscreen use, sun exposure habits, dietary vitamin D, or non-prescription supplement use were not available and would benefit future studies. Essentially, the study design is cross-sectional, with inherent limitations, so an analysis of longitudinal data of future Waves will be important to identify the determinants of vitamin D deficiency. 

## 5. Conclusions

In conclusion, the high prevalence of vitamin D deficiency identified in this cohort highlights the need to raise awareness and have adequate public health strategies for achieving optimum vitamin D concentrations to support successful ageing. Based on our findings, there is a particular need to raise awareness and target those most at risk of vitamin D deficiency. At a population level, supplement uptake remains low and a mandatory food fortification policy needs to be urgently considered as it may have the widest reach in tackling vitamin D deficiency, since it has shown to be enormously successful and safe in other far northern European countries [33].

## Figures and Tables

**Figure 1 nutrients-11-01253-f001:**
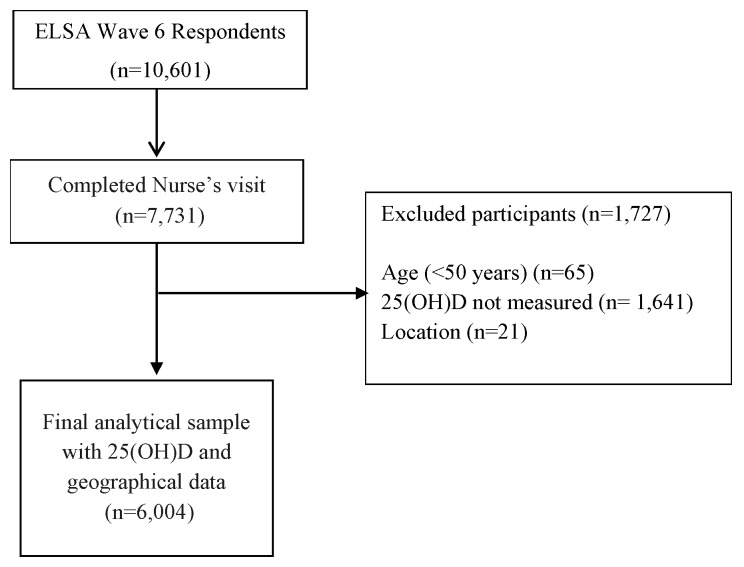
Study design. ELSA, English Longitudinal Study of Ageing.

**Figure 2 nutrients-11-01253-f002:**
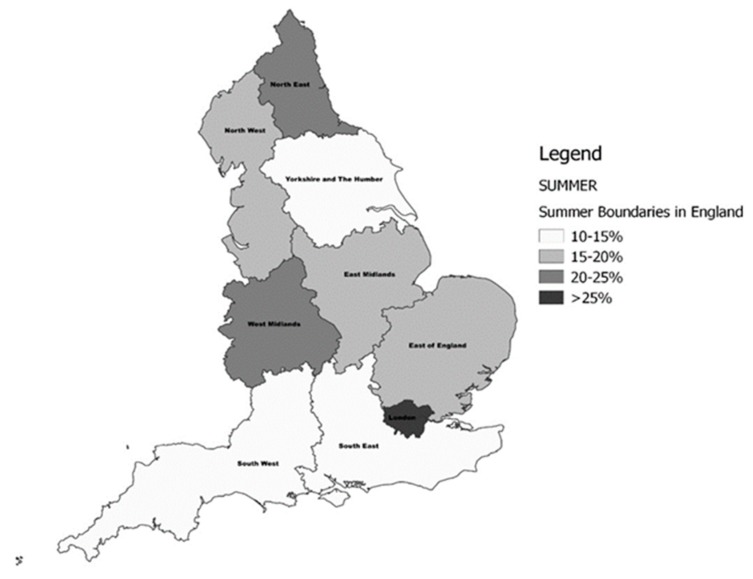
The prevalence of vitamin D deficiency (Institute of Medicine (IOM) <30 nmol/L) in English regions during the Summer period.

**Figure 3 nutrients-11-01253-f003:**
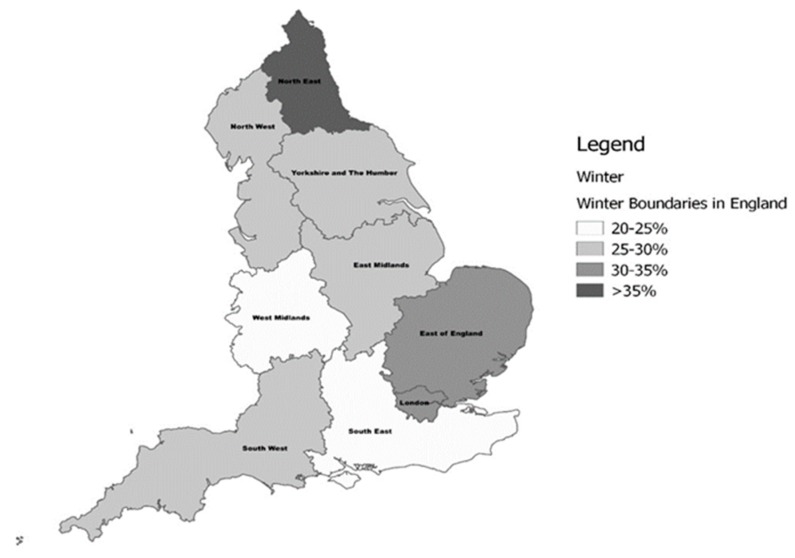
The prevalence of vitamin D deficiency (IOM <30 nmol/L) in English regions during the Winter period.

**Table 1 nutrients-11-01253-t001:** Participant characteristics stratified by region of residence in England (*n* = 6004).

Variables	North (*n* = 1692)	Midlands (*n* = 2036)	South (*n* = 2276)	All (*n* = 6004)
	n (%) ± SD	n (%) ± SD	n (%) ± SD	n (%) ± SD
**Age, years**	66.3 ± 8.8	66.6 ± 8.9	66.3 ± 8.8	66.4 ± 8.8
**Female**	950 (56.2)	1114 (54.7)	1227 (53.9)	3291 (54.8)
**Education** ≥O-level	1183 (69.9)	1420 (69.7)	1789 (78.6) ***	4392 (73.2%)
No qualification	429 (25.4)	521 (25.6)	390 (17.1) ***	1340 (22.4)
**Marital Status**				
^a^ Married	1111 (65.7)	1357 (66.7)	1566 (68.8) *	3997 (66.6)
Single	99 (5.9)	116 (5.7)	168 (7.4) *	320 (5.3)
Widow	250 (14.8)	270 (13.3)	279 (12.3) *	763 (12.7)
**Health and lifestyle factors**				
**BMI (kg/m²)**	28.3 ± 5.1	28.2 ± 4.9	27.7 ± 4.8 ***	28.0 ± 4.9
Obese	541 (32.7)	601 (30.7)	589 (26.6) ***	1731 (29.7)
**Physical Activity**				
Vigorous (>1/wk)	376 (22.2)	451 (22.2)	532 (23.4)	1359 (22.6)
Moderate (>1/wk)	1055 (62.4)	1332 (65.4)	1546 (67.9) ***	3933 (65.6)
**Current smoker**	205 (12.1)	244 (11.9)	246 (10.8)	695 (11.6)
**Alcohol 5–6 days/week**	101 (5.9)	99 (4.8)	156 (6.9)	356 (5.9)
**SR Health**				
Excellent	189 (11.2)	256 (12.6)	327 (14.4) **	772 (12.9)
Poor	114 (6.7)	137 (6.7)	109 (4.8) **	360 (6.0)
^b^ **Limiting illness**	576 (34.0)	668 (32.8)	643 (28.3) ***	1887 (31.4)
**Polypharmacy**	474 (28.0)	538 (26.4)	533 (23.4) ***	1546 (25.7)
**Predictors of Vitamin D Status**				
^c^ Winter blood sample	1011 (59.7)	1219 (59.9)	1334 (58.6)	3564 (59.4)
^d^ Sun holiday travel	953 (56.3)	1200 (58.9)	1304 (57.3)	3457 (57.6)
VitD supp user	84 (4.9)	85 (4.2)	93 (4.1)	262 (4.4)
**25(OH)D**, nmol/L	47.4 ± 23.0	48.1 ± 23.9	50.0 ± 23.2 **	48.7 ± 23.4

Notes: SD, standard deviation; SR, self-reported; wk, week; winter, October to March. ^a^ Married includes those married for the first time, those who had remarried, and those who had a legally recognized civil partnership. ^b^ if self-reported long-standing illness is limiting. ^c^ blood sample taken during the winter season. ^d^ sun holiday in last 12 months. ANOVA comparison between North and Midlands using South as the reference group. P value denoting significant levels between groups *p* < 0.05 *, *p* < 0.01 **, *p* < 0.001 ***.

**Table 2 nutrients-11-01253-t002:** Prevalence of vitamin D deficiency, stratified by age and sex (*n* = 6004).

**IOM Deficiency < 30 nmol/L (*n* = 1423)**				
**%**	**50–59 (*n* = 469)**	**60–69 (*n* = 492)**	**70–79 (*n* = 325)**	**80+ (*n* = 137)**
Overall	27.4 *	20.4 ***	22.2 ***	32.0
Males	28.1	19.3	17.3	26.3
Female	26.9	21.5	26.1	36.8
^a^ Season-Winter	32.9	24.3	27.8	35.6
**ES Deficiency <50 nmol/L (*n* = 3317)**				
**%**	**50–59 (*n* = 1007)**	**60–69 (*n* = 1251)**	**70–79 (*n* = 785)**	**80+ (*n* = 273)**
Overall	58.9	52.1 ***	53.6 ***	63.8
Males	59.0	51.8	50.1	58.2
Female	58.9	52.3	56.4	68.4
^a^ Season-Winter	66.7	58.7	60.3	69.1
**Prevalence of vitamin D deficiency, stratified by age, among D supplement users ^b^ (*n* = 262)**				
N (%)	50–59 (*n* = 33)	60–69 (*n* = 105)	70–79 (*n* = 90)	80+ (*n* = 34)
<30 nmol/L (*n* = 27)	9 (27.3)	8 (7.6)	8 (8.9)	2 (5.9)
<50 nmol/L (*n* = 72)	15 (45.4)	25 (23.8)	19 (21.1)	13 (38.2)

Notes: IOM, Institute of Medicine; ES, Endocrine Society. ^a^ Winter defined by extended vitamin D calendar October to April. ^b^ Reported taking prescribed Vitamin D supplement for osteoporosis treatment. P value denoting significant levels between groups *p* < 0.05 *, *p* < 0.001 ***.

**Table 3 nutrients-11-01253-t003:** Weighted logistic regression for predictors of vitamin D deficiency (<30 nmol/L) in ELSA study participants (*n* = 6004).

	Unadjusted Model 1		Adjusted Model 1	
Demographic Variables	OR	(95%CI)	OR	(95%CI)
**Female**	1.21 **	(1.04, 1.41)	1.23 **	(1.04, 1.44)
**Age**				
50–59 (Reference)	[1]		[1]	
60–69	0.71 ***	(0.59, 0.84)	0.85	(0.69, 1.05)
70–79	0.79 *	(0.66, 0.97)	0.89	(0.68, 1.16)
80+	1.31 *	(1.01, 1.69)	1.42 *	(1.01, 1.96)

**Non-white ethnicity**	3.59 ***	(2.4, 5.4)	3.8 ***	(2.39, 6.05)
**Marital Status**—Widow	1.59 ***	(1.31, 1.93)	1.44 ***	(1.15, 1.79)
Single	1.37 *	(1.04, 1.82)	1.33	(0.98, 1.81)
**Employment**—retired	0.79 **	(0.69, 0.92)	0.79 *	(0.65, 0.95)
**Region**—North	[1]		[1]	
Midlands	0.93	(0.78, 1.11)	0.95	(0.79, 1.14)
South	0.79 *	(0.66, 0.96)	0.78 **	(0.64, 0.95)
^a^**Latitude** (°N)	1.11 ***	(1.05, 1.17)	1.11 ***	(1.04, 1.17)
**Modifiable Health and Lifestyle Factors**				
**BMI**—Normal	0.73 ***	(0.61, 0.87)	0.81 *	(0.67, 1.00)
Obese	1.59 ***	(1.36, 1.86)	1.32 **	(1.09, 1.58)
**Current smoker**	2.18 ***	(1.78, 2.67)	1.88 ***	(1.51, 2.34)
**SR health**—Excellent	[1]		[1]	
Poor	2.26 **	(1.67, 3.06)	1.49 *	(1.08, 2.07)
Fair	3.49 ***	(2.43, 5.02)	1.99 ***	(1.33, 2.96)
**VitD supplement use**	0.39 ***	(0.25, 0.64)	0.28 ***	(0.17, 0.45)
**Season** (summer)	0.52 ***	(0.45, 0.61)	0.47 ***	(0.40, 0.56)
**Sun travel**	0.54 ***	(0.47, 0.63)	0.74 ***	(0.63, 0.86)
**PA**-Vigorous (>1/week)	0.46 ***	(0.37, 0.57)	0.68 ***	(0.55, 0.86)
Moderate (>1/week)	0.50 ***	(0.43, 0.58)	0.74 ***	(0.62, 0.88)

Notes: OR, odds ratios; CI, confidence interval; SR, self-reported; PA, physical activity. For the adjusted model, variables included socio-demographics (age in years, gender, and region of residence), health-related behaviors (current smoker, vitamin D supplement use), physical factors (vigorous PA > 1/week, BMI), and season (winter/summer blood sampling). All variables were included and assessed using the same categorization/groupings; for visual presentation only, categories that reached statistical significance are shown. ^a^ Replaced region in the adjusted model. P value denoting significant levels between groups *p* < 0.05 *, *p* < 0.01 **, *p* < 0.001 ***.

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
