# Peer review of "The Prevalence and Determinants of Vitamin D Status in Community-Dwelling Older Adults: Results from the English Longitudinal Study of Ageing (ELSA)"

_nutrients, 2019, doi:10.3390/nu11061253_

Reviewer 1 Report

The manuscript is well written and the analyses are sound. In my opinion, the findings with respect to latitude in England are very interesting and have high implications for public health. I only have one minor comment. Because many readers only read the abstract, there is a problem of selctive reporting. The authors only reported the negative determinants of vitamin D deficiency and not the positive determinants (Factors such as being retired, regular vigorous physical activity, vitamin D supplement use, sun travel and summer season were significant positive correlates of vitamin D deficiency.) When I read the abstract, I thought that something must have gone wrong because low physical activity was not listed among the determinants of vitamin D deficiency. To avoid such misunderstandings, please report all significant derminants (positive or negative) of vitamin D deficiency/insufficiency in the abstract.

Author Response

Review Report From – Reviewer 1

The manuscript is well written, and the analyses are sound. In my opinion, the findings with respect to latitude in England are very interesting and have high implications for public health. I only have one minor comment. Because many readers only read the abstract, there is a problem of selective reporting. The authors only reported the negative determinants of vitamin D deficiency and not the positive determinants (Factors such as being retired, regular vigorous physical activity, vitamin D supplement use, sun travel and summer season were significant positive correlates of vitamin D deficiency.) When I read the abstract, I thought that something must have gone wrong because low physical activity was not listed among the determinants of vitamin D deficiency. To avoid such misunderstandings, please report all significant determinants (positive or negative) of vitamin D deficiency/insufficiency in the abstract.

 Response to Reviewer’s comments

 We thank Reviewer 1 for their time in reviewing the manuscript and for providing valuable comments. We have now added more comprehensive information to the abstract on all determinants of vitamin D (positive and negative).

___________________________

Reviewer 2 Report

The manuscript investigated the prevalence and determinants of vitamin D status in community-dwelling older adults. It employed a national sample with 6004 participants. The study design was appropriate and the results were presented clearly with reasonable discussions. This study is significant for the health of elderly,specifically their bone health and preventing falls or fractures. The study findings will provide valuable information for the ageing health policy, especially for the north areas.

Only one small comment: please ensure the full names were provided for the abbreviations at the first appearance, e.g. IOM in the abstract, HSE in the line 67. 

Author Response

Thank you for the comments - we have made the changes as suggested. 

Reviewer 3 Report

Here, the authors describe the results of a cross-sectional analysis of the wave 6 survey of the ELSA cohort, a study of England-dwelling older adults (DOB<=Feb 1952), with a goal to assess the frequency of vitamin D deficiency and determinants thereof. Using two cutoffs of vitamin D deficiency (<30nmol/L, <50nmol/L), the authors found prevalence estimates of deficiency of 26.4% and 58.7%, respectively. Key determinants identified included female sex, older age (80+), smoking, non-White ethnicity, higher BMI, and self-reported health, as well as a notable latitudinal gradient despite the relatively small latitudinal range of England.

Overall this was a good use of data and adds to the literature regarding this important topic. I don’t think it’s all that novel, as such, but it is necessary and useful data that will inform policy and practice in this population. I have various comments and edits, as below and in the annotated copy. Provided these issues are remedied, I’d be happy to review a revised version.

Main comments

Some of the analysis methods employed could be a bit clearer. In particular, I was unclear as to the multivariable model used, this not being very clearly explained in the Methods, nor specified in the relevant tables. Thus, I would suggest the authors clarify this in both places, in particular enumerating the covariates adjusted for in the Tables. I was particularly concerned that all the variables enumerated in Supp Tables 3 and 4 were included in the model, as this would not be appropriate for reasons of collinearity, among others.

Further to this, I didn’t find Figure 4 to be that helpful. I’d suggest pulling Supplemental Tables 3 and 4 into the main paper instead of being supplemental. If table/figure limits is an issue, the authors could possibly consolidate Figures 2 and 3 and drop Figure 4.

Regarding analysis methods, though logistic regression is commonly employed for analyses of dichotomous outcome terms like those used here, given the cross-sectional nature of the study sample, logistic regression is inappropriate and is liable to yield inflated measures of association. Instead, log-binomial regression should be used. In STATA, this can be done using the glm command, specifying a binomial (or Poisson with robust standard errors if binomial won’t converge) and a log link. This is generally comparable to the same operation for logistic regression (glm with binomial family and a logit link) but estimates a prevalence ratio rather than an odds ratio.

The weighted regression is not clear as to what was done. The authors note in passing this was done to account for biases in sampling, but further detail is needed. I was concerned when noting the marked drop in sample size between the parent sample (n=10601) and the analytical sample (6004). Typically one would approach this by a transparent adjustment approach, examining the characteristics of those with and without data and potentially adjusting for those covariates found to significantly differ between groups. I am not averse to a weighting approach if it accomplishes the same ends but only desire that the methods used be more clearly spelled out.

There are various instances in the tables where categorical terms, for instance marital status, employment status, or self-reported health are not fully presented in the tables. For instance, between Supplemental Tables 3 and 4, these parameters have different numbers of strata presented. I am unclear whether those omitted were because they were not significant or if these variables were categorised/consolidated differently between models. I really hope the latter is not the case as this would make comparing results between models impossible. I would suggest the authors be consistent in presenting all the data or, if they do find it necessary to consolidate/drop strata of variables, to do so consistently between models, and to be clear in the Methods what was done and why.

In some instances, the authors seem to use unexpected reference levels for their categorical terms. For instance, I think they’re using overweight as the reference for BMI? This is not typical but not strictly speaking, wrong. I would just suggest being clear in the tables as to what the reference group is.

Though perhaps not central to this paper, I found it interesting that the participants in the South region have significantly better health parameters, including BMI, physical activity, self-rated health, and limiting illnesses. What do the authors think may account for this? Is this described in another ELSA paper and if so, is a similar regional variability seen? This, along with the higher proportions with educational qualifications makes me wonder if this is particularly influenced by London being in this region? Differential participation bias, in this case healthy participant bias, is a potential issue that might be considered and which may complicate comparability between regions.

In the Results describing the prevalence estimates of vitamin D deficiency by the two cutpoints, there appears to be a marked difference in the proportions with deficiency by the 50nmol/L cutoff between overall (57.3%) and restricted to those 60+yo (22.5%). The authors say the proportions are similar but this is rather different, persisting on weighting it seems. Can the authors speak to this?

The authors mention that London did not follow the latitudinal gradient in line with the other regions, but don’t elaborate. This isn’t all that big an issue, as it’s in keeping with the population aggregation effect typical for the big cities, where people more in need of medical care or with chronic conditions are more likely to live closer to bigger cities where needed services are located. This should be elaborated upon, however, both to clarify how London doesn’t follow the pattern in the Results, as well as explaining the population aggregation effect in the Discussion.

Other comments:

Abstract

-          The author state that assessment of vitamin D deficiency at higher latitudes is less explored, but then proceed to enumerate several studies in the Discussion.

-          Define IOM at first usage.

-          Specify OR (or PR) for female sex and age 80+ to be consistent.

-          Be consistent in presentation, showing results for both IOM and ES-defined deficiency, or alternately just stating that “similar results were seen for the other cutoff” or some such.

-          As with the body text, be clear what model covariates are being adjusted for. It is not enough to say “important predictors of vitamin D status”.

-          Specify latitude cutoff or range defining “south of England”.

Introduction

-          Refs 3-6 don’t really match up with vitamin D in diet.

-          Some brief explanation of the place of 25(OH)D as the main circulating and diagnostic metabolite of vitamin D would be good.

-          Consider verb choice for the discussion of the cross-sectional results of ref 13. One can’t refer to an active temporality (increasing/decreasing with age), but can only say that levels were higher/lower at different ages.

Methods

-          Define HSE.

-          A minor point but how is it that anyone in this study could have an age under 50 if sampling was in 2012 and they had to have a DOB of at least Feb 1952? Wouldn’t this make them at least 59? Or are these instances where the DOB was just incorrectly recorded?

-          Clarify that those 21 excluded for Location reasons are those who specified their residence as outside England and who, I presume, were just visiting at the time of study?

-          The comment about extreme outliers of 25(OH)D is superfluous, given as no one reached it.

-          Regarding vitamin D supplementation, the authors only include prescribed supplementation with calcium. The authors note in the Discussion that over the counter supplementation is infrequent in the older population but this would run at odds with the Australian experience. Is it not legal in England or just capped at a low dosage or? Also, is it common for doctors to prescribe only vitamin D or is it solely vitamin D plus calcium? My thoughts go to patients with autoimmune diseases like MS, where vitamin D supplementation is quite common throughout the life course and thus, wonder whether there might be other non-osteoporosis usage.

-          Clarify for we international readers what O-level entails for education. Is this secondary school or?

-          Given as the 8ft gait speed test was only in a subgroup, and as this doesn’t seem to be presented in the Results, I think this could be omitted.

-          The Covariates section of the Results is a bit poorly organised and could benefit from some restructuring for clarity.

-          Prevalence is not a rate!

Results/tables/figures

-          For Table 1, please add a column for all persons.

-          For Table 1, please add a row for mean (SD; range) 25(OH)D.

-          The superscript symbols in Table 1 don’t line up with the abc in the subcaption, though this is used for Supplemental Table 1. I imagine this is vestigial from a previous iteration but should be fixed.

-          In Table 1, as well as variously in the text, clarify what Winter blood implies. I presume this means the samples were taken in the winter period but just need to clarify this.

-          In Table 2, clarify that the bottom section is Prevalence of vitamin D deficiency, stratified by age and restricted to vitamin D supplement users.

-          As previous, I think Figure 4 can be cut.

-          In Supplemental Table 1, add columns for the non-deficient levels for each cutoff.

-          In Supplemental Tables, ensure model covariates for adjusted models are specified.

Discussion

-          Capitalise The in the TILDA.

Formatting/other

-          Consider word choices. I’ve circled various such instances in the annotated copy.

-          Various minor typographical and grammatical errors should be remedied.

-          Maintain consistency of number of significant figures in tables.

-          When the authors refer to gender, I believe they mean sex.

-          When the authors use the term, multivariate, I believe they mean multivariable. Multivariate refers to multiple outcomes whereas multivariable refers to multiple determinants.

Author Response

Here, the authors describe the results of a cross-sectional analysis of the wave 6 survey of the ELSA cohort, a study of England-dwelling older adults (DOB<=Feb 1952), with a goal to assess the frequency of vitamin D deficiency and determinants thereof. Using two cut-offs of vitamin D deficiency (<30nmol/L, <50nmol/L), the authors found prevalence estimates of deficiency of 26.4% and 58.7%, respectively. Key determinants identified included female sex, older age (80+), smoking, non-White ethnicity, higher BMI, and self-reported health, as well as a notable latitudinal gradient despite the relatively small latitudinal range of England.

Overall this was a good use of data and adds to the literature regarding this important topic. I don’t think it’s all that novel, as such, but it is necessary and useful data that will inform policy and practice in this population. I have various comments and edits, as below and in the annotated copy. Provided these issues are remedied, I’d be happy to review a revised version.

Main comments

1. Some of the analysis methods employed could be a bit clearer. In particular, I was unclear as to the multivariable model used, this not being very clearly explained in the Methods, nor specified in the relevant tables. Thus, I would suggest the authors clarify this in both places, in particular enumerating the covariates adjusted for in the Tables. I was particularly concerned that all the variables enumerated in Supp Tables 3 and 4 were included in the model, as this would not be appropriate for reasons of collinearity, among others.

 Response: Thank you for your comments, we have added this information under the relevant tables and edited the methods section in the manuscript to reflect this. We applied MANOVA to all predictors to test the assumptions of the model, before including them in a stepwise regression. Regarding variables included in the model, only variables that reached statistical significance in the initial analysis (Supplemental Tables 3 and 4) were included in the models and collinearity was assessed by scatter plots and Variance Inflation Factors.

2. Further to this, I didn’t find Figure 4 to be that helpful. I’d suggest pulling Supplemental Tables 3 and 4 into the main paper instead of being supplemental. If table/figure limits is an issue, the authors could possibly consolidate Figures 2 and 3 and drop Figure 4.

Response:

We are happy to move supplementary Tables 3 and 4 into the main paper as suggested. We have now formatted the main manuscript to include both Tables 3 and 4 and revised numbering of tables and citations in the text to mirror this as appropriate.

 We have retained Figure 4, but as a Supplementary Figure only, since it may be helpful in communicating the findings to a lay audience or in presentations.  

 3. Regarding analysis methods, though logistic regression is commonly employed for analyses of dichotomous outcome terms like those used here, given the cross-sectional nature of the study sample, logistic regression is inappropriate and is liable to yield inflated measures of association. Instead, log-binomial regression should be used. In STATA, this can be done using the glm command, specifying a binomial (or Poisson with robust standard errors if binomial won’t converge) and a log link. This is generally comparable to the same operation for logistic regression (glm with binomial family and a logit link) but estimates a prevalence ratio rather than an odds ratio.

 Response: We agree that the cross-sectional nature of the data is, in itself, an important limitation, which we acknowledge (discussion). Logistic regression analysis was employed in the present study as it is considered an acceptable analytical approach and is widely used. We appreciate that log-binomial regression is an alternative analysis strategy, that some researchers prefer. We applied the same analysis strategy as recently used in The Irish Longitudinal Study in Ageing (TILDA) that identified factors associated with vitamin D deficiency. Both ELSA and TILDA employ similar study designs, comparable variables, that allow comparison of findings. Indeed, our findings support similar factors, but the present study (ELSA) identified ethnicity, since TILDA was not ethnically diverse.

Action taken: Based on the comments here, we have now clarified the points raised regarding our logistical regressing models and analysis in the manuscript, which we hope are now clearer and more helpful to the reader. Furthermore, we have re-emphasised the limitations around cross-sectional data and analysis in the discussion section.

 4. The weighted regression is not clear as to what was done. The authors note in passing this was done to account for biases in sampling, but further detail is needed. I was concerned when noting the marked drop in sample size between the parent sample (n=10601) and the analytical sample (6004). Typically, one would approach this by a transparent adjustment approach, examining the characteristics of those with and without data and potentially adjusting for those covariates found to significantly differ between groups. I am not averse to a weighting approach if it accomplishes the same ends but only desire that the methods used be more clearly spelled out.

 Response: The core (analytical) sample for 25(OH)D is based on ELSA participants that took part in the nurse visit, and thus had a blood sample. Response to the blood sample was modelled on a range of household- and individual-level information collected from the ELSA wave 6 main interviews. Significant differences in participant characteristics were noted between those who engaged in the nurse visits and subsequently provided a blood sample, compared to those who did not respond to a nurse’s visit. To bring the characteristics more in line with what is known about the study population, available cross-sectional blood weights were applied to the multivariate logistic regression. The final blood sample weight was a product of this non-response weight and the wave 6 nurse weight.  Detailed descriptions of the weighting principles applied by the ELSA Study can be found in the technical reports available at www.ifs.org.uk/elsa

Action:  the above information and clarification has been added to the manuscript statistics section to explain the points related to blood weights.

 5. There are various instances in the tables where categorical terms, for instance marital status, employment status, or self-reported health are not fully presented in the tables. For instance, between Supplemental Tables 3 and 4, these parameters have different numbers of strata presented. I am unclear whether those omitted were because they were not significant or if these variables were categorised/consolidated differently between models. I really hope the latter is not the case as this would make comparing results between models impossible. I would suggest the authors be consistent in presenting all the data or, if they do find it necessary to consolidate/drop strata of variables, to do so consistently between models, and to be clear.

Response: We apologise for this initial confusion. Only the statistically significant findings / categories were presented. However, all variables were included and assessed based on the same categorisation/groupings. We have clarified this in Tables.

6. In some instances, the authors seem to use unexpected reference levels for their categorical terms. For instance, I think they’re using overweight as the reference for BMI? This is not typical but not strictly speaking, wrong. I would just suggest being clear in the tables as to what the reference group is.

Response: The reference group was underweight participants and not overweight and we have clarified this in the results section.

7. Though perhaps not central to this paper, I found it interesting that the participants in the South region have significantly better health parameters, including BMI, physical activity, self-rated health, and limiting illnesses. What do the authors think may account for this? Is this described in another ELSA paper and if so, is a similar regional variability seen? This, along with the higher proportions with educational qualifications makes me wonder if this is particularly influenced by London being in this region? Differential participation bias, in this case healthy participant bias, is a potential issue that might be considered and which may complicate comparability between regions.

 Response: The regional demographic / lifestyle differences are known, the previous HSE study (pre-dates ELSA) demonstrated similar. The South of England is more affluent than the more industrialised North, so education and SES. London, interestingly, did not follow the trend for lower vitamin D deficiency in the rest of the Southern Region (addressed later). Within ELSA, the area of London contributed a relatively small sample (n=483) with a higher level of vitamin D deficiency noted overall and during the Summer period. Unfortunately, we have insufficient information to explain this, and no information on the specific location areas within London. It may reflect urban lifestyles, and a heterogeneous population in that it may reflect lower SES, poverty. It is extremely interesting, but unfortunately beyond the scope of this paper.

8. In the Results describing the prevalence estimates of vitamin D deficiency by the two cutpoints, there appears to be a marked difference in the proportions with deficiency by the 50nmol/L cutoff between overall (57.3%) and restricted to those 60+yo (22.5%). The authors say the proportions are similar but this is rather different, persisting on weighting it seems. Can the authors speak to this?

Response: We thank the reviewer for highlighting this error, and have amended the text to reflect this ‘For older age categories of ≥60 years and ≥65 years, vitamin D deficiency (<30nmol/L) and deficiency (<50nmol/L) were similar at 22.6% and 22.5% (weighted: 25.3% and 25.3%) and 54.2% and 53.9% (weighted: 56.1% and 55.9%) respectively’.

9. The authors mention that London did not follow the latitudinal gradient in line with the other regions, but don’t elaborate. This isn’t all that big an issue, as it’s in keeping with the population aggregation effect typical for the big cities, where people more in need of medical care or with chronic conditions are more likely to live closer to bigger cities where needed services are located. This should be elaborated upon, however, both to clarify how London doesn’t follow the pattern in the Results, as well as explaining the population aggregation effect in the Discussion.

Response: London did not follow the same trend as the other southern regions, in contrast higher vitamin D deficiency and lower serum 25(OH)D than in the South. This finding may reflect urban lifestyles or a heterogeneous population with extremes of affluence and poverty, or a population aggregation, where people more in need of medical care or with chronic conditions aggregate in cities. It is worth noting, also, that within ELSA, London contributed a relatively small sample (n=483).

Action: We already have an existing section in the Discussion regarding London, we have expanded this to include the points raised on population aggregation.

We now also state ‘how’ London deviates from the pattern in the results.

 Other comments:

Abstract –

The author state that assessment of vitamin D deficiency at higher latitudes is less explored, but then proceed to enumerate several studies in the Discussion.  

 Response: We have qualified this statement in the abstract, and now state ‘in ageing’.

1.Define IOM at first usage.

Response: This has been defined.

 2. Specify OR (or PR) for female sex and age 80+ to be consistent.

Response: This has now been edited

 3. Be consistent in presentation, showing results for both IOM and ES-defined deficiency, or alternately just stating that “similar results were seen for the other cut-off” or some such.

Response: This has now been changed.

 4. As with the body text, be clear what model covariates are being adjusted for. It is not enough to say “important predictors of vitamin D status”.

Response: These details have now been added in though we are restricted by the word count of the abstract.

 5. Specify latitude cut-off or range defining “south of England”.

Response: These details have now been added in the abstract.

Introduction

6. Refs 3-6 don’t really match up with vitamin D in diet.

Response: Some of the references are there to support the statements that physical activity and high quality diet are important for successful aging. We have added in 2 different references for dietary vitamin D as per the reviewer’s suggestion.

7. Some brief explanation of the place of 25(OH)D as the main circulating and diagnostic metabolite of vitamin D would be good.

Response: This has now been added to the manuscript.

8. Consider verb choice for the discussion of the cross-sectional results of ref 13. One can’t refer to an active temporality (increasing/decreasing with age), but can only say that levels were higher/lower at different ages.

Response: We agree with the reviewer and this has now been changed in the manuscript.

 Methods

9. Define HSE

Response: This has now been defined

10. A minor point but how is it that anyone in this study could have an age under 50 if sampling was in 2012 and they had to have a DOB of at least Feb 1952? Wouldn’t this make them at least 59? Or are these instances where the DOB was just incorrectly recorded?  

Response: Participants aged<50 were partners of the core members who have been invited for interview and included after Wave 1.  Noted in methods

11. Clarify that those 21 excluded for Location reasons are those who specified their residence as outside England and who, I presume, were just visiting at the time of study?  

Response: The 21 participants excluded for locations reasons resided outside of England, specifically Scotland and Wales. Noted in methods.

 12. The comment about extreme outliers of 25(OH)D is superfluous, given as no one reached it.

Response: We feel this is important information for the reader that we considered this and the fact that no one in the study population actually reached this high level.

13. Regarding vitamin D supplementation, the authors only include prescribed supplementation with calcium. The authors note in the Discussion that over the counter supplementation is infrequent in the older population but this would run at odds with the Australian experience. Is it not legal in England or just capped at a low dosage or? Also, is it common for doctors to prescribe only vitamin D or is it solely vitamin D plus calcium? My thoughts go to patients with autoimmune diseases like MS, where vitamin D supplementation is quite common throughout the life course and thus, wonder whether there might be other non-osteoporosis usage.

Response: In the ELSA dataset, the only information available on vitamin D supplement usage was under the medication category, i.e. the prescribed drugs list (WTC codes). The category was described as vitamin D and calcium for osteoporosis. Hence the usage is low, but similar to TILDA (around 8.5%). Unfortunately, ELSA did not capture specific information on nutritional supplements or over-the-counter vitamin D use. 

14. Clarify for we international readers what O-level entails for education. Is this secondary school or?

Response: This detail has now been added to the manuscript.

15. Given as the 8ft gait speed test was only in a subgroup, and as this doesn’t seem to be presented in the Results, I think this could be omitted.

Response: Thank you for highlighting, we have removed this from the manuscript.

The Covariates section of the Results is a bit poorly organised and could benefit from some restructuring for clarity.

Response: We have restructured and re-written the paragraph to clarify.

 Prevalence is not a rate 

Response: This has been edited where appropriated in the paper

 Results/tables/figures

For Table 1, please add a column for all persons.

Response:   a new column has been added to Table 1 as requested

For Table 1, please add a row for mean (SD; range) 25(OH)D.   

Response: This has now been captured in Table 1 as requested

The superscript symbols in Table 1 don’t line up with the abc in the subcaption, though this is used for Supplemental Table 1. I imagine this is vestigial from a previous iteration but should be fixed.    

Response: This has been revised as suggested

In Table 1, as well as variously in the text, clarify what Winter blood implies. I presume this means the samples were taken in the winter period but just need to clarify this.  

Response: noted and amended

In Table 2, clarify that the bottom section is Prevalence of vitamin D deficiency, stratified by age and restricted to vitamin D supplement users.

Response: amended to indicate that this analysis is among supplement users only – the n value, (sub-sample size), is also provided 

As previous, I think Figure 4 can be cut.

Response:  As outlined earlier, we have removed from the main paper but retained as a Supplemental Figure.

In Supplemental Table 1, add columns for the non-deficient levels for each cut-off.

Response:  Our goal was to present the data consistently across tables to capture vitamin D deficiency by 2 cut-offs and show corresponding data for ‘All’ participant for the reader’s reference.  We stayed away from presenting additional data on non-deficient to avoid confusion and excessive  data,  since the focus of this study, as per the aims, was strongly on vitamin D deficiency [ie, to establish the prevalence of serum 25(OH)D deficiency in older adults, ii) to determine the prevalence of deficiency by region of residence]. We have included sufficiency in past studies mores in exploring 25(OH)D concentrations  e.g. quartiles, or categories for 50/75/100 nmol/l. If the Reviewer feels this is valuable information for the paper, we can add it to follow, our concern was on keeping to vitamin D deficiency.

In Supplemental Tables, ensure model covariates for adjusted models are specified.

Response:  This information had been added to relevant tables

 Discussion

Capitalise ‘The’ in the TILDA.

 Response: This has now been done.

 Formatting/other

Consider word choices. I’ve circled various such instances in the annotated copy.

Various minor typographical and grammatical errors should be remedied.

Maintain consistency of number of significant figures in tables.

Response:  This has been revised for formatting issues

When the authors refer to gender, I believe they mean sex.

Response: Though the ELSA study design and methods paper refers to gender. We have revised the manuscript to the term sex as advised.

 When the authors use the term, multivariate, I believe they mean multivariable. Multivariate refers to multiple outcomes whereas multivariable refers to multiple determinants. Response: Replaced in the paper, in line with Reviewer’s comment

Round  2

Reviewer 3 Report

My thanks to the authors for their prompt and industrious response to my comments. I am generally quite satisfied with the revised version. My only additional comment is, if the authors want to retain Supp Figure 4, they need to include explanation of what the exposure categories are in the caption.

Otherwise, this is very well done and I’m happy for it to proceed.